# N-Heterocyclic carbene-catalyzed enantioselective synthesis of planar-chiral cyclophanes via dynamic kinetic resolution

Jiayan Li[1,4], Ziyang Dong [1,4], Yang Chen[2], Zhanhui Yang [2], Xinen Yan[1], Meng Wang[3], Chenyang Li [3] ✉ & Changgui Zhao [1] ✉

Planar-chiral cyclophanes have gained considerable concerns for drug discovery due to their unique conformational strain and 3D structure. However, the enantioselective synthesis of planar-chiral cyclophanes is a long-standing challenge for the synthetic community. We herein describe an N-heterocyclic carbene (NHC)-catalyzed asymmetric construction of planar-chiral cyclophanes. This transformation occurs through a dynamic kinetic resolution (DKR) process to convert racemic substrates into planar-chiral macrocycle scaffolds in good to high yields with high to excellent enantioselectivities. The *ansa* chain length and aromatic ring substituent size is crucial to achieve the DKR approach. Controlled experiments and DFT calculations were performed to clarify the DKR process.

Macrocycles exhibit unique properties including shape diversity, conformational pre-organization, and conformational flexibility, compared to small-sized rings, particularly, five- to seven-membered rings[1,2]. The large surface area and tunable conformation of macrocycles increase the likelihood that the macrocycle will make meaningful contact with a biological target[1,3]. The compilation of macrocyclic small molecule screening libraries is critical to success when challenging, underexploited, or poorly "druggable" biological targets, such as lorlatinib, are involved[1,4]. Cyclophanes, which are molecules with an aromatic scaffold bearing a cross-linked chain, also called an *ansa* chain, are a subset of macrocycles[5–7]. Notably, cyclophane with a short *ansa* chain and bulky aromatic substituents exhibit planar chirality, which arises from the restricted bond flip of the aromatic ring[8–10]. Although planar-chiral cyclophanes are part of small molecule screening collections, these compounds have been increasingly used in drug development[11,12], asymmetric synthesis[13], and functional materials[14] (Fig. 1A), and additional planar-chiral cyclophanes could benefit these collections. However, the enantioselective synthesis of planar-chiral cyclophanes represents a formidable challenge to the synthetic community and hampers their widespread inclusion in drug development efforts[15,16].

Historically, the asymmetric synthesis of cyclophanes relied on chiral pools and chiral auxiliaries[17–21]. Asymmetric catalysis now provides access to the enantiopure planar-chiral cyclophanes[16], as seen in Tanaka's seminal asymmetric arene formation strategy to produce chiral metacyclophanes via rhodium-catalyzed alkyne cyclotrimerization[22]. Inspired by Tanaka's pioneering work, several elegant enantioselective inter- and intramolecular [2 + 2 + 2] cycloaddition processes were reported for planar-chiral cyclophane assembly (Fig. 1B, I)[23–25]. In addition, asymmetric macrocyclization strategies have also provided access to planar-chiral cyclophanes[26–32]. Very recently, Collins[28], Yang[29], and Li[30] independently developed enzyme-, chiral phosphoric acid (CPA)- or Pd-catalyzed asymmetric macrocyclization to achieve planar-chiral macrocycles (Fig. 1B, II). In addition to these processes, the desymmetrization is also an attractive method to afford planar-chiral structures. Shibata achieved the enantioselective synthesis of planar-chiral cyclophanes by *ortho*-lithiation of the achiral cyclophanes (Fig. 1B, III)[33]. The fourth is dynamic kinetic resolution (DKR) of achiral cyclophanes (Fig. 1B, IV). The challenges in generating planar-chiral cyclophanes by a DKR process involve identifying appropriately sized aromatic ring substituents and *ansa* chain lengths, which will allow rapid racemization of the substrate but limit the bond flip of the aromatic ring in the

[1]Key Laboratory of Radiopharmaceuticals, Ministry of Education, College of Chemistry, Beijing Normal University, Beijing 100875, China. [2]Department of Organic Chemistry, College of Chemistry, Beijing University of Chemical Technology, Beijing 100029, China. [3]Key Laboratory of Theoretical and Computational Photochemistry, Ministry of Education, College of Chemistry, Beijing Normal University, Beijing 100875, China. [4]These authors contributed equally: Jiayan Li, Ziyang Dong. ✉e-mail: chenyang.li@bnu.edu.cn; cgzhao@bnu.edu.cn

**A** Selected examples of molecules containing a planar-chiral cyclophane.

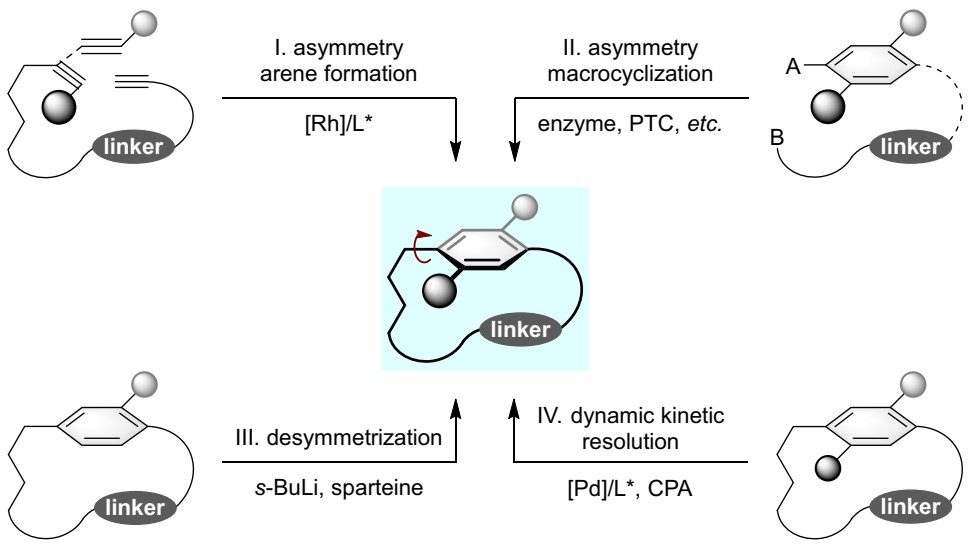

Cyclindrocyclophane A                FVIIa inhibitor                Lorlatinib

**B** Approaches for the catalytic enantioselective synthesis of cyclophanes.

I. asymmetry arene formation [Rh]/L*

II. asymmetry macrocyclization enzyme, PTC, *etc.*

III. desymmetrization *s*-BuLi, sparteine

IV. dynamic kinetic resolution [Pd]/L*, CPA

**C** Design of an NHC-catalyzed enantioselective DKR of cyclophanes.

**This design**

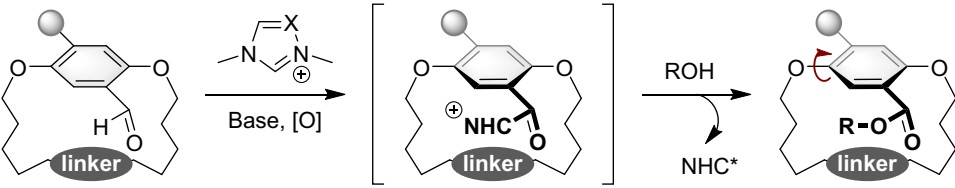

Base, [O]          ROH          NHC*

**Fig. 1 | Importance of cyclophanes and the enantioselective synthesis. A** Selected examples of molecules containing a planar-chiral cyclophane. **B** Approaches for the catalytic enantioselective synthesis of cyclophanes. **C** Design of an NHC-catalyzed enantioselective DKR of cyclophanes.

product. Both components are necessary to ensure the configurational stability of the resulting planar-chiral cyclophanes. In this scenario, Shibata was able to produce the chiral cyclophanes by a double asymmetric Sonogashira coupling of achiral [n, n]paracyclophanes[34]. More recently, Yang demonstrated an elegant CPA-catalyzed asymmetric electrophilic aromatic amination protocol to deliver planar-chiral macrocycles[35].

Despite the previously mentioned straightforward approaches to afford planar-chiral cyclophanes, limitations include variable

enantiomeric excess for different substrates, low reaction efficiencies, limited scope, and the use of stoichiometric amounts of chiral reagents. Further, the current protocols are restricted to a handful of catalytic models, and organocatalytic enantioselective synthetic routes are still in their infancy[16]. Consequently, these challenges prompted us to develop a DKR approach to achieve optically pure planar-chiral cyclophanes.

Chiral N-heterocyclic carbenes (NHCs) have been widely used as powerful organo-catalysts to access molecular architectures[36–44].

**Fig. 2 | Optimization of the reaction conditions.** [a]Reaction conditions: A mixture of **1a** (0.10 mmol), **2a** (0.12 mmol), DQ (0.10 mmol), Base (10 mol%), and NHC catalyst (10 mol%) in solvent (2.0 mL) was stirred at room temperature for 24 h, the number in the black circle is the *ansa* chain length. [b]Isolated yield. [c]Determined by HPLC using a chiral stationary phase. [d]NHC catalyst (5 mol%); the reaction was stirred for 48 h. [e]The reaction was performed at 40 °C.

| Entry[a] | NHC | Base | Solvent | Yield (%)[b] | er (%)[c] |
|---|---|---|---|---|---|
| 1 | I | DBU | THF | 87 | 42:58 |
| 2 | II | DBU | THF | 97 | 64:36 |
| 3 | III | DBU | THF | 92 | 50:50 |
| 4 | IV | DBU | THF | 78 | 93:7 |
| 5 | V | DBU | THF | 70 | 93:7 |
| 6 | VI | DBU | THF | 90 | 65:35 |
| 7 | VII | DBU | THF | 80 | 16:84 |
| 8 | IV | K₂CO₃ | THF | 10 | 90:10 |
| 9 | IV | Cs₂CO₃ | THF | 69 | 93:7 |
| 10 | IV | KO^tBu | THF | 44 | 91:9 |
| 11 | IV | DBU | EtOAc | 61 | 87:13 |
| 12 | IV | DBU | 2-Me-THF | 66 | 93:7 |
| 13 | IV | DBU | MTBE | 63 | 87:13 |
| 14[d] | V | DBU | THF | 76 | 93:7 |
| 15[e] | V | DBU | THF | 76 | 95:5 |

However, most previous reports focused on the synthesis of central chirality, and more recently, axial chirality[45–49]. Only recently has successful control of planar-chiral ferrocenes through NHC catalysis been reported by Chi[50]. Herein, we disclose an NHC-catalyzed DKR of racemic cyclophanes by an oxidative esterification reaction to afford the planar-chiral cyclophanes in good to high yields with high to excellent enantioselectivities (Fig. 1C). We have also demonstrated that our enantioenriched products can be easily converted into diverse planar-chiral scaffolds for potential library inclusion and asymmetric catalysis through cross-coupling reactions. Controlled experiments and DFT calculations were performed to clarify the DKR process.

## Results

### Reaction optimization

We set out to explore the DKR of cyclophanes by examining the reaction of [14]paracyclophane **1a** with (3,5-di-tert-butylphenyl) methanol **2a** in the presence of triazolium NHC as catalyst (Fig. 2). The atropisomerism of **1a** was observed by variable temperature ¹H NMR experiments (below −50 °C), suggesting a low rotational energy barrier for [14]cyclophane **1a** at room temperature. Using 1,8-diazabicyclo[5.4.0]undec-7-ene (DBU) as base, 3,3′,5,5′-tetratert-butyldiphenoquinone (DQ) as oxidant, and THF as solvent, various chiral triazoline NHC catalysts **I**–**VII** were initially screened (entries 1–7). The pyrrolotriazolium-derived NHC catalyst **I** exhibited high conversion

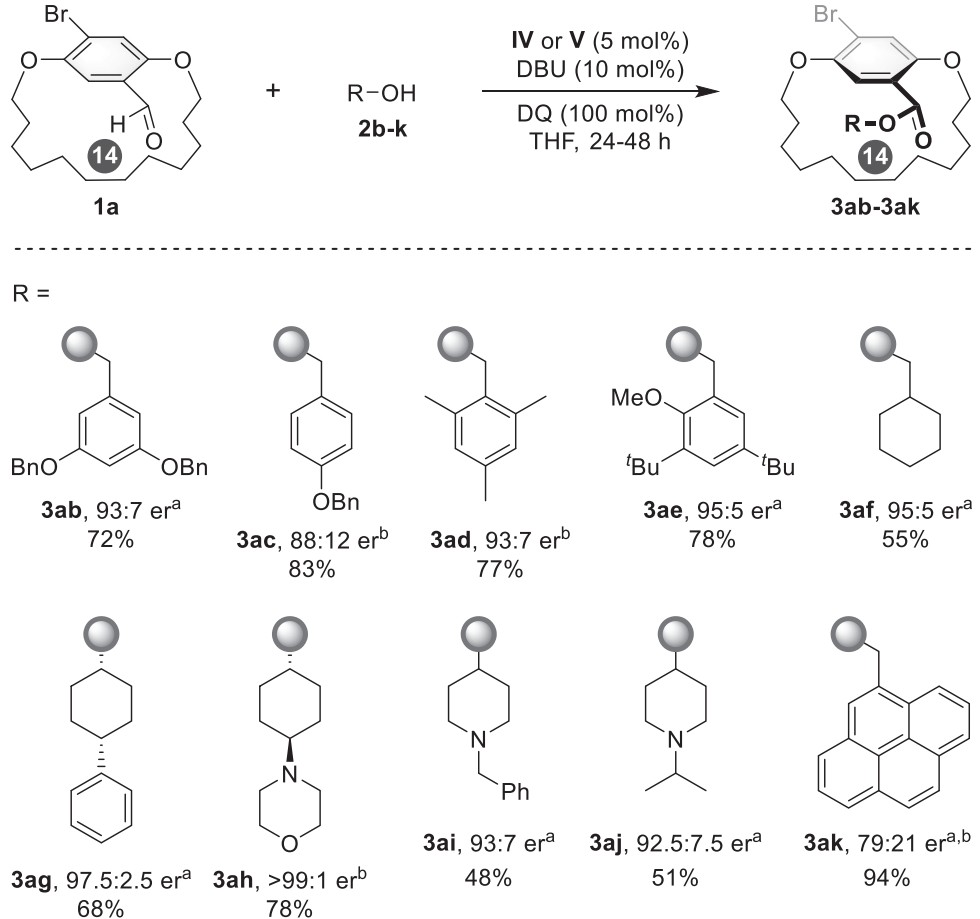

**Fig. 3 | Evaluation of the effect of various alcohol substrates.** [a]The reaction was performed at room temperature, **IV** was used as catalyst. [b]The reaction was performed at 40 °C, **V** was used as catalyst.

but low enantioselectivity, while amino alcohol-derived chiral NHC **VII** afforded the desired product in good yield with moderate enantioselectivity (entries 1 and 7). The N-substituents of the indanol-derived catalysts have an obvious impact on the enantioselectivity of the reaction, as N-2,4,6-$(Me)_3C_6H_2$ (Mes), N-2,4,6-$(Br)_3C_6H_2$, or N-$C_6F_5$ substituents gave the product in high yield but with poor enantioselectivity (entries 2, 3, and 6). Pleasingly, the enantioselectivity was dramatically increased when bulky N-2,4,6-$(^iPr)_3C_6H_2$ or N-2,4,6-$Cy_3C_6H_2$[51] substituted indanol-derived NHC **IV** or **V** catalyst was used (entry 4–5, 93:7 er). Further base optimization indicated that strong bases, such as DBU and $Cs_2CO_3$, resulted in higher yields compared to relatively weak bases (entries 8–10). Subsequent solvent experimentation demonstrated that THF achieved the highest yields (entries 11–13). Decreasing the catalyst loading was not detrimental to reaction outcomes (entry 14). Interestingly, increase of temperature yielded the product in a higher er (entry 15). The improved enantioselectivity might arise from a faster racemization rate of the substrate, which facilitate the DKR process. The absolute configuration of **3aa** was unambiguously determined by X-ray diffraction analysis of its derivative (*vide infra*).

## Substrate scope

With the optimized conditions in hand, we then evaluated the effect of the alcohol, starting with substituents on the phenyl ring of the benzyl alcohol. Regardless of the steric and electronics of the aromatic ring substituents, the desired planar-chiral cyclophanes (**3ab**–**3ad**) were afforded good yield with moderate to good enantioselectivities (Fig. 3). When substrates bearing bulkier groups were used, good yields and high enantioselectivities were achieved (**3ae**, 78% yield and

95:5 er). When other cyclic secondary alcohols were evaluated, the corresponding products **3af**–**3aj** were obtained in 48–78% yield with er ranging from 93:7 – >99:1. Moreover, the benzyl alcohol with a 4-prenyl group was not an appropriate substrate for this reaction under the optimized conditions (**3ak**).

The investigation of *ansa* chain length and substituent size was conducted using **2e** as model substrate (Fig. 4). Reducing to 11- or 12-membered *ansa* chains, the reaction went through a kinetic resolution process, in which planar-chiral products **3be** and **3ce** were obtained in 86:14 er and 91.5:8.5 er, respectively, while **1c** was recovered with 51% yield and 82:18 er. Using a 13-membered *ansa* chain was feasible, delivering the planar-chiral [13]paracyclophane **3de** with moderate yield and good enantioselectivity. Increasing the *ansa* chain length and using a 15-membered compound was tolerated and gave the product **3ee** in 72% yield with a 95:5 er value, while [16]paracyclophane **3fe** led to the loss of planar chirality due to the relative low rotation barrier.

In addition, an examination of the aromatic ring substituent size was performed (Fig. 4). 4-H substituted [14]paracyclophane resulted in the loss of planar chirality (**3ge**), but [14]paracyclophanes bearing chloro, methyl, or iodo groups on the benzene ring allowed the NHC-catalyzed DKR to occur, generating the planar-chiral cyclophanes in good yields and enantioselectivities (**3he**–**3je**). Installment of phenyl or ethynyl groups on the aromatic ring was also tolerated. The products **3ke** and **3le** were obtained in high er value, and corresponding substrates **2k** and **2l** were recovered in low er value, which suggested racemization of the substrates occurred. These results indicate that the DKR of planar-chiral [n]paracyclophanes is highly dependent on *ansa* chain length and the size of substituents in the aromatic ring,

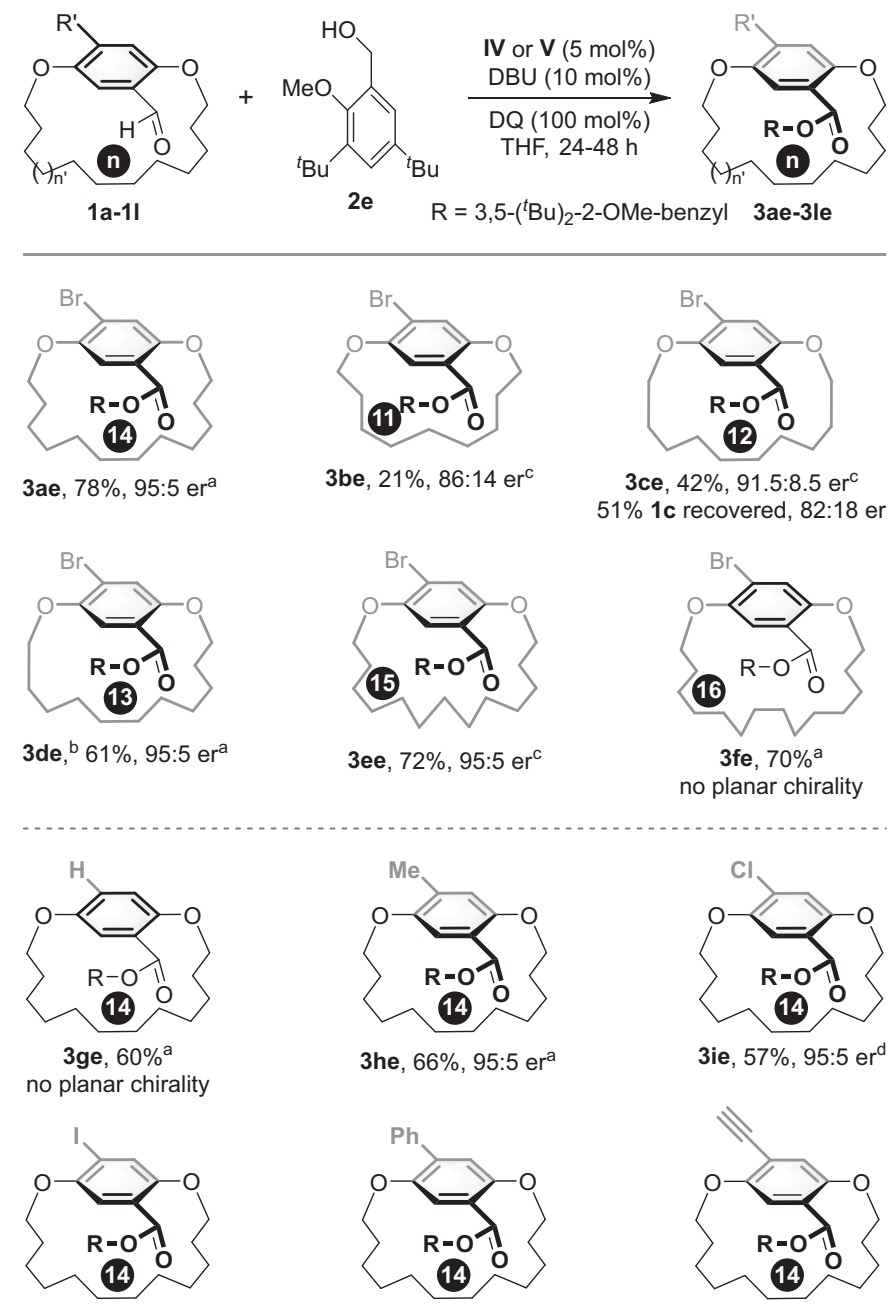

**Fig. 4 | Evaluation of the ansa chain length and aromatic ring substituent size.** [a]The reaction was performed at room temperature, **IV** was used as catalyst. [b]The reaction was performed at 40 °C, **V** was used as catalyst. [c]0.2 mmol of **2a** was used. [d]The reaction was performed at 50 °C, **V** was used as catalyst. [e]The reaction was stirred at room temperature for 24 h, **1k** was recovered in 55:45 er, and **1 l** was recovered in 50:50 er.

which allows substrate racemization but limits bond flip in the product's aromatic ring.

Encouraged by these results, we continued to explore the scope of [n]paracyclophanes in this NHC-catalyzed DKR reaction. A diverse array of achiral paracyclophanes with a variety of *ansa* chain lengths and substituents were tested (Fig. 5). [13]paracyclophane with a chloro group at the 4-position of the phenyl ring performed well to deliver the corresponding planar chiral product (**5ae**) in good yield and enantioselectivity. [14] and [15]paracyclophanes bearing different substitutes (vinyl, iodo, ethynyl, phenyl) at the phenyl ring were also accommodated and provided the enantioenriched products (**5be–5fe**). Heteroaromatic rings like 3-thienyl could be introduced at the 4-position of the substrates (**5ge, 5he**). [16] and [17]paracyclophane with 2-naphthyl,

benzofuran-6-yl or benzo[b]thiophen-3-yl yielded the produts in high enantioselectivity (**5ie–5ke**), as could [18]paracyclophane with benzofuran-6-yl albeit with lower enantioselectivity (**5le**). In addition, modifications of the *ansa* chain were also investigated. The yield of [14] and [15]paracyclophane with an ester group or nitrogen-linked *ansa* chain decreased to 26% and 16% (**5me, 5ne**), respectively. The decreased yield of **5me** probably originated from substrate decomposition under the reaction conditions. [15]paracyclophane with an oxygen-linked *ansa* chain afforded the product in good yield with high enantioselectivity (**5oe**). To assess the configurational stability of the planar-chiral products, **3ae** in toluene was heated to 110 °C. Notably, HPLC analysis indicated that no racemization of **3ae** was observed, even after 7 days at this temperature (see Supplementary Information).

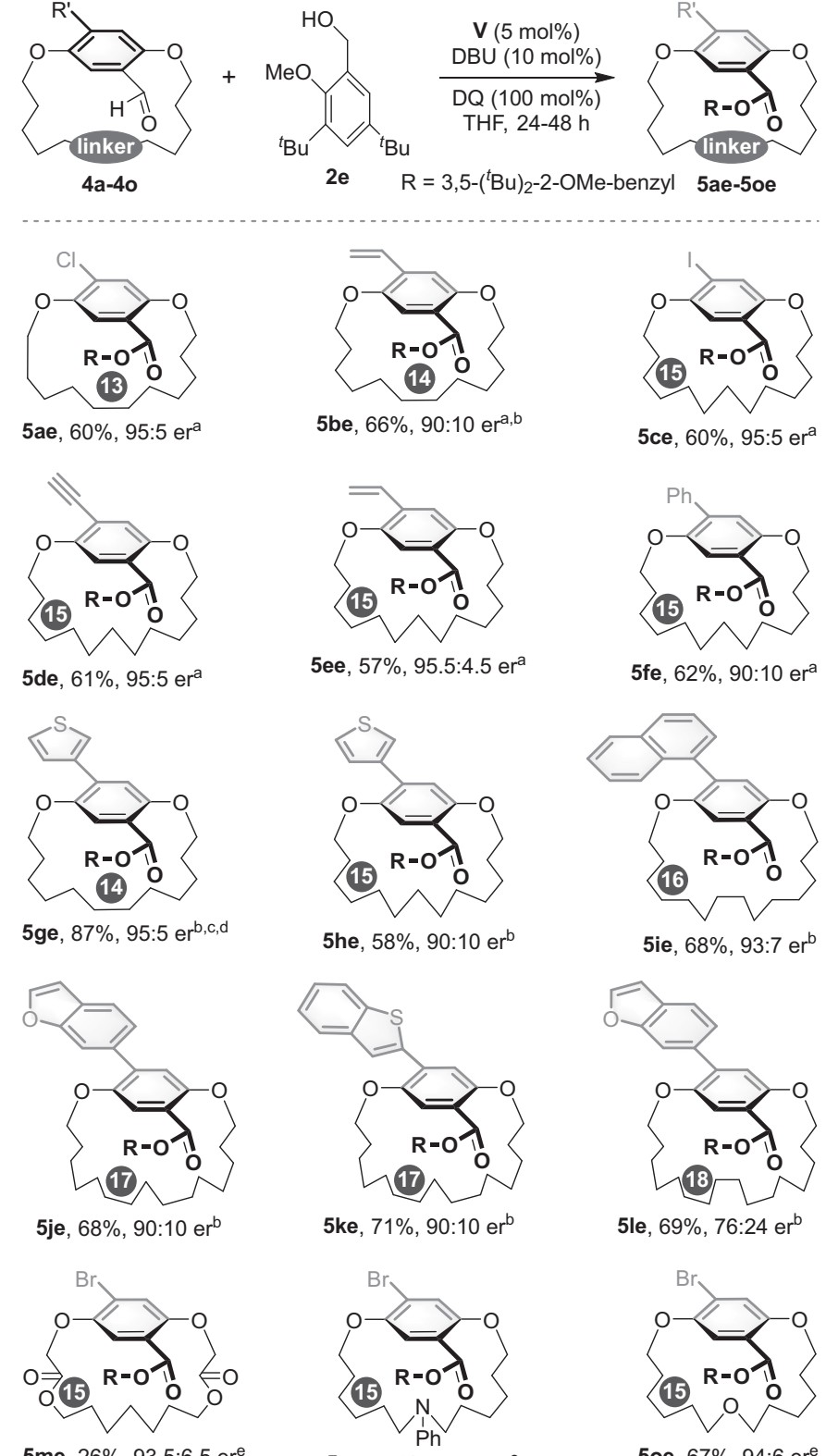

**Fig. 5 | The scope of [n]paracyclophanes.** [a]The reaction was performed at 40 °C. [b]The reaction was performed at 50 °C. [c]The reaction was performed at room temperature, **5ge** was obtained in 86:14 er. [d]The reaction was performed at 40 °C, **5ge** was obtained in 92:8 er. [e]The reaction was performed at 40 °C, **IV** was used as catalyst.

To expand the potential utility of this process, we further manipulated the planar-chiral cyclophane (Fig. 6A). The ester in **3ae** could be reduced with LiAlH$_4$ to generate benzylic alcohol **6** in 98% yield with only a slight erosion of enantioselectivity. In addition, a Ni-catalyzed cross-coupling reaction of **3ae** with diphenylphosphine afforded planar-chiral phosphine **7** in 61% yield with a slight loss of stereochemical integrity due to the vigorous reaction conditions. Moreover, the enantiomerically enriched triazole **8** was obtained by a

**A** Synthetic transformation of enantioenriched [14]paracyclophane **3ae**.

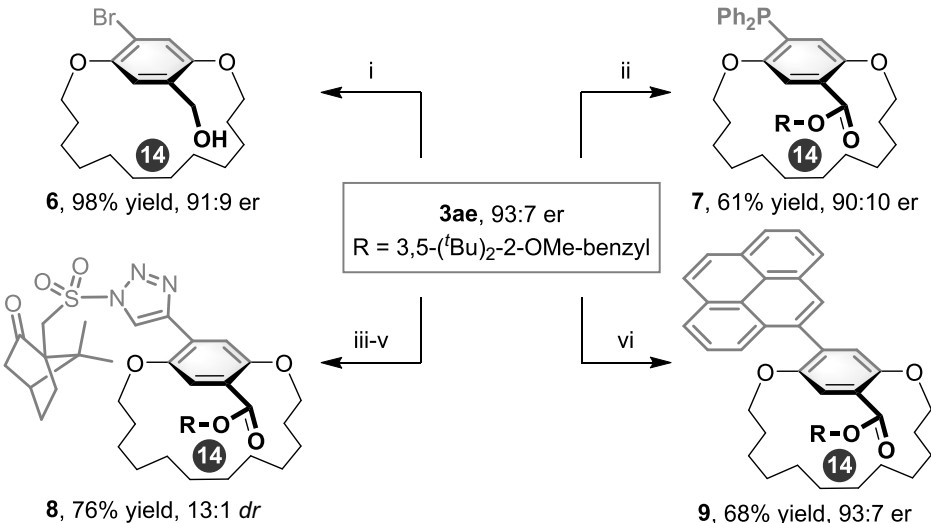

**B** X-ray of planar-chiral cyclophane **10**.

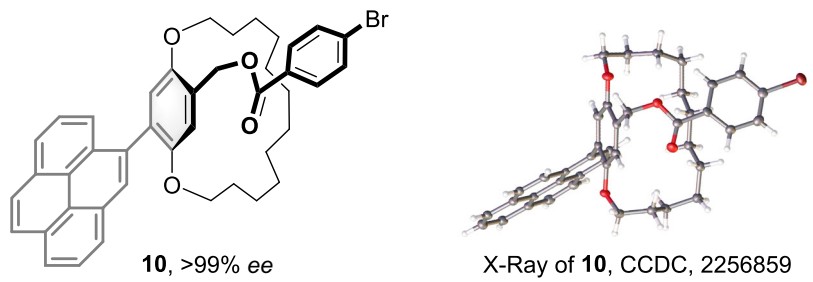

**10**, >99% *ee*

X-Ray of **10**, CCDC, 2256859

**Fig. 6 | Derivatizations of the chiral product. A** Synthetic transformation of enantioenriched [14]paracyclophane **3ae**. Reaction conditions: (i) LiAlH₄, 0 °C, THF, 2 h. (ii) Ni(dppe)Cl₂, Ph₂PH, Et₃N, DMF, 120 °C, 12 h. (iii) Pd(PPh₃)₂Cl₂, CuI, TMS-acetylene, Et₃N, 50 °C, 10 h, THF, 91% yield, 93:7 er. (iv) TBAF·3H₂O, THF, rt, 4 h, 87% yield, 93:7 er. (v) CuTC, (*R*)-Camphor-10-sulfonyl azide, toluene, rt, 4 h, 96% yield, 13:1 dr. (vi) Pd(PPh₃)₄, Na₂CO₃, pyren-1-ylboronic acid, 1,4-dioxane, 100 °C, 24 h, 68% yield, 93:7 er. **B** X-ray of planar-chiral cyclophane **10**.

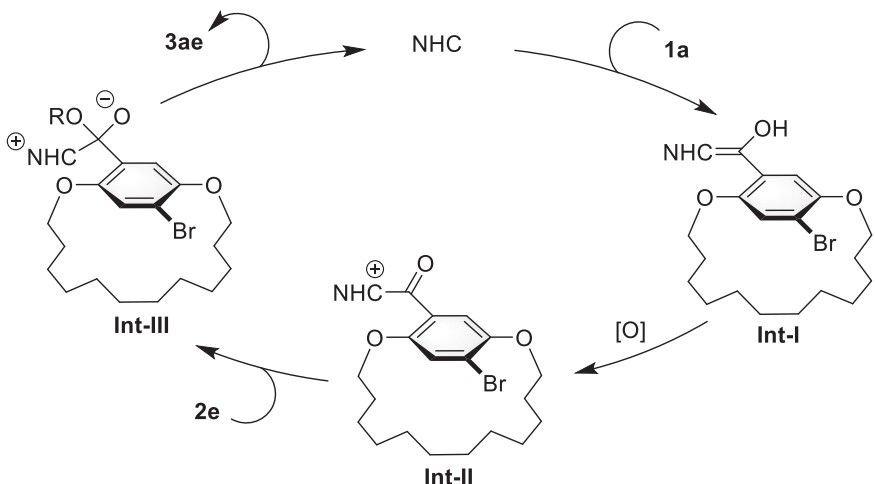

**Fig. 7 | Postulated mechanism of the reaction.**

Sonogashira cross–coupling, desilication, and CuTC-catalyzed click reaction sequence. Finally, 3ae was easily transferred to corresponding pyrene-bearing product **9** in 68% yield and 93:7 er through a Suzuki–Miyara cross–coupling reaction with pyren-1-ylboronic acid. The absolute configuration of **10** was unambiguously determined by

X-ray diffraction analysis (https://www.ccdc.cam.ac.uk/data_request/cif) (Fig. 6B).

A postulated mechanism is illustrated in Fig. 7. The reaction starts from the addition of NHC catalyst **IV** to aldehyde **1a** to generate Breslow intermediate **Int-I**. Then, the oxidation of **Int-I** forms acylazolium

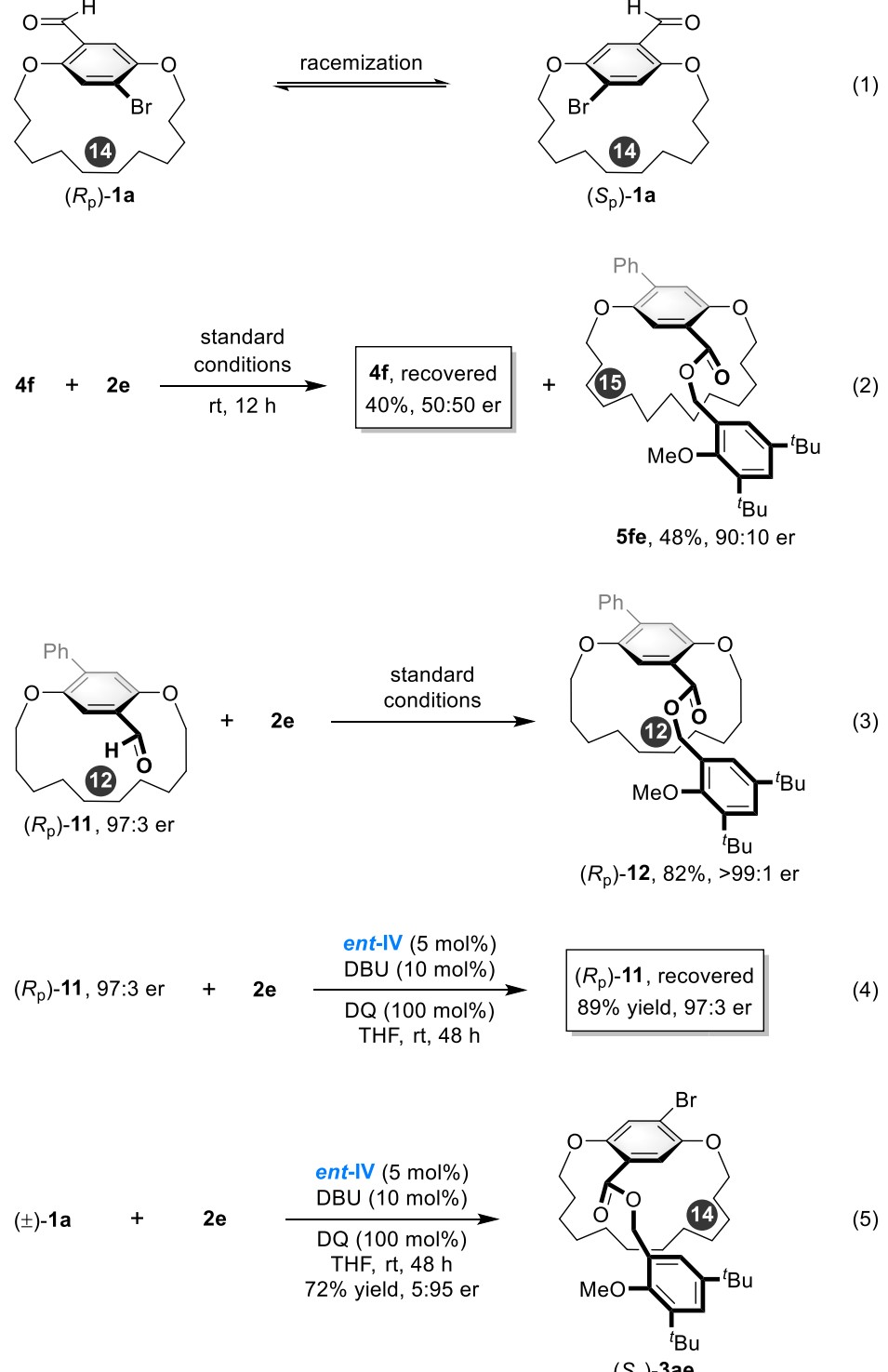

**Fig. 8 | Controlled experiments of the dynamic kinetic resolution process.**

intermediate **Int-II**. Finally, the esterification of **Int-II** with benzylic alcohol **2e** delivers the planar-chiral product **3ae** and releases the free NHC catalyst **IV**.

The DKR process originates from the rapid bond flip of the aromatic ring in the two enantiomers ([$S_p$]−**1a** and [$R_p$]−**1a**) and the reaction-rate difference with the addition of NHC catalyst (Fig. 8, eq 1). To clarify the DKR process, several controlled experiments were conducted. The reaction of [15]paracyclophane **4f** with **2e** was performed under standard conditions for 12 h; **5fe** was obtained in 48% yield with

90:10 er, and **4f** was recovered as racemates in 40% yield. The result indicates that the racemization of **4k** is rapid and not the rate-determined step (Fig. 8, eq 2). In addition, as illustrated in the investigation of substrate scope, [12]paracyclophane underwent kinetic resolution, thus, enantioenriched 4-phenyl-substituted [12]paracyclophane ($R_p$)−**11** was treated under the optimized reaction conditions, the reaction proceeds smoothly to deliver ($R_p$)−**12** in 82% yield with >99:1 er value (Fig. 8, eq 3). Alternatively, only a trace amount of ($R_p$)−**12** was obtained by using **ent-IV** as NHC catalyst (<5% yield), and

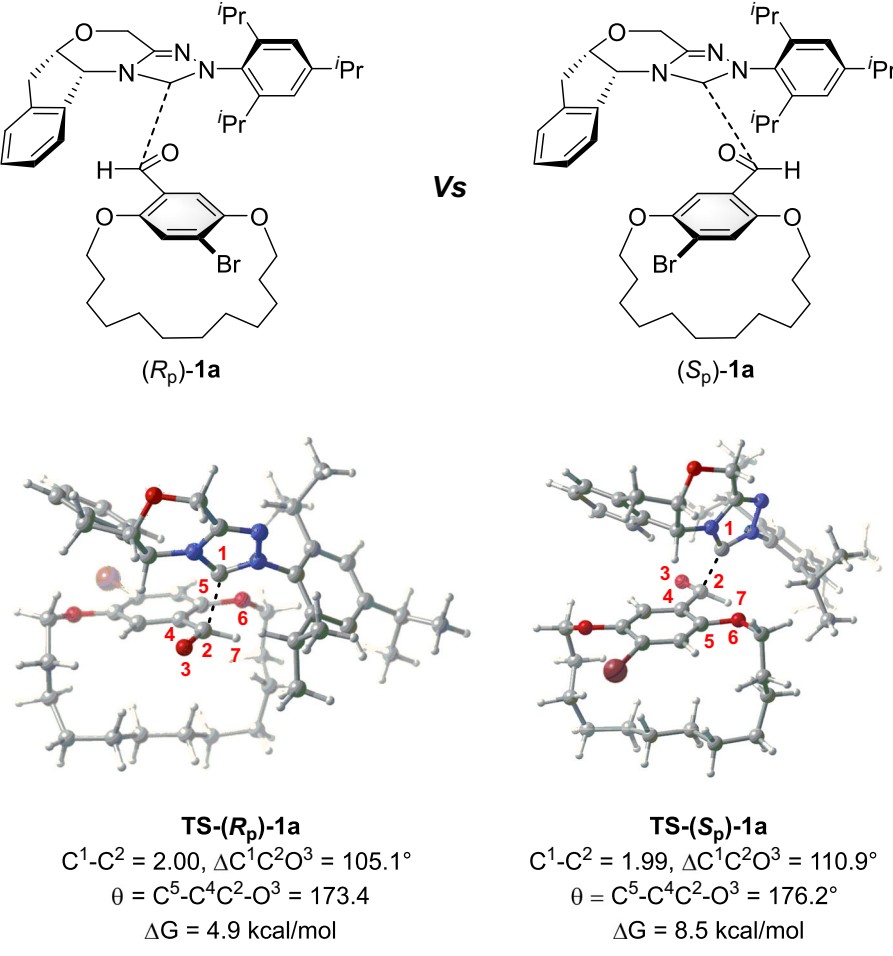

**Fig. 9 | Calculated energy profile.** Calculated energy profile of the addition of NHC catalyst **IV** to [14]paracyclophane **1a**.

the ($R_p$)−**11** was recovered in 89% yield with 97:3 er value (Fig. 8, eq 4). Further, the reaction of **1a** and **2e** with ***ent*-IV** as catalyst gave a reverse selectivity, suggesting that the ($R_p$)−**11** was unmatched with ***ent*-IV** (Fig. 8, eq 5). These results indicate that the reaction of NHC catalyst **IV** to ($R_p$)-**11** is kinetically favored.

To further explore the energy-difference for the addition of NHC catalyst **IV** to the enantiomers ($S_p$)-**1a** and ($R_p$)-**1a**, a density functional theory (DFT) study was performed using Gaussian 09. As illustrated in Fig. 9, the energy barrier of the addition of NHC catalyst **IV** to ($R_p$)-**1a** is 3.6 kcal/mol lower than that of ($S_p$)-**1a**; thus, the addition of NHC catalyst **IV** to ($R_p$)-**1a** is kinetically favored. Further analysis of transition states **TS-($S_p$)-1a** and **TS-($R_p$)-1a** using atoms-in-molecules[52,53] and noncovalent interactions analyses[54] reveal that the weak interactions in **TS-($R_p$)-1a** are stronger than those observed in **TS-($S_p$)-1a**, which leads to **TS-($R_p$)-1a** having a lower energy barrier. Interactions in **TS-($R_p$)-1a** include: seven C−H...π interactions, four C−H...O hydrogen bond interactions, two C−H...N hydrogen bond interactions, one Lp...π interaction, one C−H...Br halogen bond interaction, and ten C−H... H−C van der Waals interactions. Interactions in **TS-($S_p$)-1a** include: four C−H...π interactions, three C−H...O hydrogen bond interactions, one C−H...N hydrogen bond interaction, two C−H...Br halogen bond interactions, and nine C-H...H−C van der Waals (See Supplementary Fig. 2). Although the calculated results are consistent with the observed experiments, a relatively low Gibbs free energy for addition of NHC catalyst to the substrate suggests a rapid rate for this transformation. While the subsequent oxidative esterification is not the enantioselectivity-determined step, together with experimental obversion that increases the reaction temperature led to a higher

enantioselectivity. Thus, the Curtin-Hammet scenario might be operative for the DKR approach[55].

## Discussion

In summary, we developed an NHC-catalyzed enantioselective synthesis of planar-chiral cyclophanes. The reaction features a DKR process involving an aldehyde oxidation esterification and affords a wide range of planar-chiral macrocycles in good to high yields with high enantioselectivities. An investigation of *ansa* chain length and the size of aromatic ring substituents indicated that these variables were crucial to generating the planar chirality. Further, the cyclophane products could be transformed into other planar-chiral macrocyclic scaffolds by simple reactions. Controlled experiments and DFT calculations were performed to clarify the DKR process. An application of these planar-chiral scaffolds for library inclusion and asymmetric catalysis are under investigation in our lab.

## Methods

Materials. For ¹H NMR, ¹³C NMR, and high-performance liquid chromatography spectra of compounds in this manuscript, see Supplementary Figures. For details of the synthetic macrocycle substrates: diphenol (10.0 mmol) and dibromides (10.0 mmol) in DMF (25.0 mL) were slowly added to a suspension of $K_2CO_3$ (3.45 g, 25.0 mmol) and NaI (170 mg, 1.0 mmol) in DMF (150 mL) at 140 °C over 18 h. The solvent was removed under reduced pressure. The aqueous phase was extracted with EtOAc (3 × 50 mL), and the combined organic layers were washed with water and brine, and dried over $Na_2SO_4$. The solvent was removed under reduced pressure, and the residue was purified by

flash column chromatography on silica gel affording the products (yield: 10 ~ 40%).

Synthesis of **3** and **5**. To a 15 mL Schlenk tube equipped with a magnetic stirring bar was added macrocycles substrates **1** or **4** (0.10 mmol), **2** (0.4 mmol), NHC precursor **IV** or **V** (0.005 mmol), DBU (0.01 mmol) and DQ (0.1 mmol). The tube was closed with a septum, evacuated, and refilled with nitrogen (3 cycles). Then, freshly distilled THF (1.0 mL) was added to the reaction mixture and stirred for 10–72 h. Upon completion (monitored by TLC), the solvent was evaporated, and the residue was purified by silica gel column chromatography to afford the planar-chiral cyclophanes **3** or **5**.

### Reporting summary

Further information on research design is available in the Nature Portfolio Reporting Summary linked to this article.

## Data availability

The authors declare that the data supporting the findings of this study are available within the article and Supplementary Information file, or from the corresponding author upon request. The X-ray crystallographic coordinates for structures reported in this study have been deposited at the Cambridge Crystallographic Data Centre (CCDC), under deposition numbers CCDC 2256859 (for (***R**_p*)-**10**). These data can be obtained free of charge from The Cambridge Crystallographic Data Centre via www.ccdc.cam.ac.uk/data_request/cif. The full experimental details for the preparation of all new compounds, and their spectroscopic and chromatographic data generated in this study are provided in the Supplementary Information/Source Data file. All data are available from the authors upon request. Source data are provided with this paper.

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

## Acknowledgements

This project was financially supported by the National Natural Science Foundation of China (No. 22171027, 22103005), the Beijing Natural Science Foundation (No. 2212009), the Fundamental Research Funds for the Central Universities (No. 2233300007) and the High Performance Computing Platform of Beijing University of Chemical Technology (BUCT). We thank Prof. Dr. Guofu Zi (Beijing Normal University) for the X-ray diffraction analysis and Dr. Stephanie A. Blaszczyk for proofread the manuscript (Moxie Medical Writing).

## Author contributions

J.Y.L. conducted main experiments; Z.Y.D. prepared several starting materials; X.E.Y. optimized the reaction conditions and prepared several starting materials during the revision of the manuscript. Y.C.and Z.H.Y. performed part of the DFT computation; M.W. and C.Y.L. rerun the computation during the revision of the manuscript. C.G.Z. conceptualized and directed the project, and drafted the manuscript with the assistance from co-authors. All authors contributed to discussions.

## Competing interests

The authors declare no competing interests.
