## [Peer Review File · Nature Communications]

N-Heterocyclic Carbene-Catalyzed Enantioselective Synthesis of Planar-Chiral Cyclophanes via Dynamic Kinetic ResolutionReviewers' Comments:

Reviewer #1:

Remarks to the Author:

[Note from the Editor: Please see attached review report]

The manuscript by Zhao and coworkers describes a highly efficient methodology to access enantioenriched paracyclophanes via dynamic kinetic resolution. A wide range of planar chiral macrocycles with different aromatic substitutions and ansa chain lengths are produced in good yields and high enantioselectivities.

The addition of a chiral NHC to paracyclophanes containing aldehyde groups results in the dynamic kinetic resolution of the starting achiral substrate by increasing the rotational barrier energy. The enantiomerically enriched Breslow intermediate is oxidized and transformed in a bulky ester that ensures the configurational stability of the final product.

The generation of planar chirality, using achiral cyclophanes as substrates, requires a careful analysis of the rotational barrier between conformers in order to classify the process as type III or IV (Scheme 1). What authors describe as III is actually a desymmetrization process that implies the removal of one element of symmetry and requires a high rotational barrier to be efficient. The authors may be mixing concepts by attributing some of the examples described by Shibata to via III, when in fact they are DKR demetrisations. Or perhaps, it is simply that the representation is not the most appropriate. This is an important aspect that authors evidenced experimentally for certain substrates that underwent kinetic resolution instead of dynamic kinetic resolution. In this respect, a discussion explaining the initial choice of the chain length and a brief analysis of the conformational behaviour (NMR for instance) of the starting cyclophanes is missing.

To clarify the reaction mechanism the authors performed several experiments and DFT calculations. The main concerns about the conclusions drawn for these studies are:

- The 3.6 Kcal/mol energy difference between TS-(Sp)-1a and TS-(Rp)-1a is sufficient to explain the preferential formation of the enantiomer Rp, proposed by the authors as the kinetically favoured. Based in these data the authors proposed that the NHC addition is irreversible (Scheme 8). Nevertheless, the energies computed for TS-(Sp)-1a and TS-(Rp)-1a (8.5 and 4.9 Kcal/mol, respectively) are too low to explain the transformation, which is carried out at room temperature for 24-48h.
Could it be more feasible to consider that enantiomer Rp is produced under Curtin-Hammet conditions?
- The authors provide information about the calculation of a single conformation for each diastereomeric transition state. Have they calculated other possibilities? Because it is easy to imagine conformational variations of the presented transition structures. For example, the aldehyde carbonyl group can be found in two orientations, and indeed the orientation shown in Figure 2 does not correspond to the computed one in SI.

An important aspect of any experimental work is the precise description of the procedures used to confirm its feasibility. In the SI, the authors do not provide the yields of the achiral macrocycles used in the DKR. Even if they were moderate or poor, they are an indication of the feasibility of the asymmetric methodology and the efforts made to improve them.

Other comments, suggestions and typos:

- Bibliographic information regarding the synthesis of optically active cyclophanes using diastereoselective methodologies is poor (see ref. 17). Is Ref. 10 correct?. References 48 and 49 are missing.
- In Table 1, negative notation for ee ratios is confusing

- Follow the result and discussion is hard, as the structure and numbering of some substrates and alcohols do not appear explicitly in all schemes.
- In the SI there are some flaws in schemes. See, for example the preparation of 4j.

1. Responses to reviewer 1:

1.1) The manuscript by Zhao and coworkers describes a highly efficient methodology to access enantioenriched paracyclophanes via dynamic kinetic resolution. A wide range of planar chiral macrocycles with different aromatic substitutions and ansa chain lengths are produced in good yields and high enantioselectivities.

Our response:

Thanks for the positive comments from this reviewer.

1.2) The addition of a chiral NHC to paracyclophanes containing aldehyde groups results in the dynamic kinetic resolution of the starting achiral substrate by increasing the rotational barrier energy. The enantiomerically enriched Breslow intermediate is oxidized and transformed in a bulky ester that ensures the configurational stability of the final product. The generation of planar chirality, using achiral cyclophanes as substrates, requires a careful analysis of the rotational barrier between conformers in order to classify the process as type III or IV (Scheme 1). What authors describe as III is actually a desymmetrization process that implies the removal of one element of symmetry and requires a high rotational barrier to be efficient. The authors may be mixing concepts by attributing some of the examples described by Shibata to via III, when in fact they are DKR demetrisations. Or perhaps, it is simply that the representation is not the most appropriate.

Our response:

Thanks for the comments. We have reorganized Scheme 1B according to your comments.

1.3) This is an important aspect that authors evidenced experimentally for certain substrates that underwent kinetic resolution instead of dynamic kinetic resolution. In this respect, a discussion explaining the initial choice of the chain length and a brief analysis of the conformational behavior (NMR for instance) of the starting cyclophanes is missing.

Our response:

Thanks for this comment. The atropisomerism of **1a** was observed by variable temperature ¹H NMR experiments (below -50 °C), suggesting a low rotational energy barrier for [14]cyclophane **1a** at room temperature. We have added the above statement in the revised manuscript.

Dynamic ¹H NMR of **1a** (CD₃Cl)

1.4) To clarify the reaction mechanism the authors performed several experiments and DFT calculations. The main concerns about the conclusions drawn for these studies are:

(a) The 3.6 Kcal/mol energy difference between TS-(Sp)-1a and TS-(Rp)-1a is sufficient to explain the preferential formation of the enantiomer Rp, proposed by the authors as the kinetically favoured. Based in these data the authors proposed that the NHC addition is irreversible (Scheme 8). Nevertheless, the energies computed for TS-(Sp)-1a and TS-(Rp)-1a (8.5 and 4.9 Kcal/mol, respectively) are too low to explain the transformation, which is carried out at room temperature for 24-48h. Could it be more feasible to consider that

enantiomer R_p is produced under Curtin-Hammet conditions?

Our response:

We agree this comment. We have added the following statement in the revised manuscript.

“Although the calculated results are consistent with the observed experiments, a relative low Gibbs free energy for addition of NHC catalyst to the substrate suggesting a rapid rate for this transformation. While the subsequent oxidative esterification is not the enantioselectivity-determined step, together with experimental observation that increase the reaction temperature led to a higher enantioselectivity. Thus, Curtin-Hammet scenario might be operative for the DKR approach.”

(b) The authors provide information about the calculation of a single conformation for each diastereomeric transition state. Have they calculated other possibilities? Because it is easy to imagine conformational variations of the presented transition structures. For example, the aldehyde carbonyl group can be found in two orientations, and indeed the orientation shown in Figure 2 does not correspond to the computed one in SI.

Our response:

We have calculated other possibilities according to your comments. We have added these results in the revised Supporting Information.

*The corresponding energy profile is summarized in **Figure S1**. For each diastereomeric pathway, we consider two orientations of the aldehyde carbonyl group, where the energetically favored one is indicated by the suffix **-1a** while the disfavored one is labeled by **-1a'**. Comparing **1a'** to the respective **1a** pathway, changing the orientation of the aldehyde carbonyl group leads to an increase of relative Gibbs free energy by 7.2 and 1.5 kcal/mol for the R_p and S_p transition state, respectively.*

Figure S1. Calculated energy profile of the addition of NHC catalyst **IV** to [14]paracyclophane **1a**. M06-2X-D3(0)/6-311+G(d,p)/PCM (THF)//B3LYP-D3/6-31G(d,p)/PCM (THF).

1.5) An important aspect of any experimental work is the precise description of the procedures used to confirm its feasibility. In the SI, the authors do not provide the yields of the achiral macrocycles used in the DKR. Even if they were moderate or poor, they are an indication of the feasibility of the asymmetric methodology and the efforts made to improve them.

Our response:

Thanks for the comments. We have provided the yields for the synthesis of achiral macrocycles in the SI based on your suggestions.

1.6) Bibliographic information regarding the synthesis of optically active cyclophanes using diastereoselective methodologies is poor (see ref. 17). Is Ref. 10 correct?. References 48 and 49 are missing.

Our response:

Thanks for the comments. We have added the references regarding the synthesis of optically active cyclophanes using diastereoselective methodologies. We have also reorganized the references accordingly in the revised manuscript.

1.7) In Table 1, negative notation for ee ratios is confusing.

Our response:

Thanks for the comments. NHC **I** and **VI** gave the product in opposite configuration. We

have revised “-58:42, -84:16” to “42:58, 16:84” in Table 1.

1.8) Follow the result and discussion is hard, as the structure and numbering of some substrates and alcohols do not appear explicitly in all schemes.

Our response:

Thanks for this comment. We have renumbered the structure in Schemes 3 and 4. We have also added the structure of alcohol in Schemes 3, 4, 6.

1.9) In the SI there are some flaws in schemes. See, for example the preparation of 4j.

Our response:

Thanks for the comments. We have fixed these issues accordingly.

2. Responses to reviewer 2:

2.1) This paper from Zhao and co-workers presented an enantioselective synthesis of planar chiral cyclophanes via DKR or KR process using NHC-catalysed oxidative esterification. This represents a nice extension of using NHC-promoted oxidative esterification for enantioselective synthesis that has been well-established for enantioselective synthesis of central chirality, axial chirality, and even planar chiral ferrocenes via KR, DKR or desymmetrization. This work was carried out in high quality in general and the manuscript was well-written. However, I would recommend major revision of this work based on the following serious concerns, and will be happy to review a revised manuscript later.

Our response:

We are pleased to thank this reviewer for the positive comments. We have revised the manuscript based on your suggestions.

2.2) the significance of this work. The author mentioned “The challenges in generating planar chiral cyclophanes by a DKR process involve identifying appropriately sized aromatic ring substituents and ansa chain lengths, which will allow rapid racemization of the substrate but limit bond flip of the aromatic ring in the product. Both components are necessary to ensure the configurational stability of the resulting planar chiral cyclophanes.” However, these were not unsolved challenges. The CPA-catalyzed asymmetric electrophilic aromatic amination via DKR process (Scheme 1B, IV) already addressed these challenges, with a more general substrate scope and higher enantioselectivity comparing to this work. A proper description of the literature precedents is suggested for the introduction section of this manuscript.

Our response:

Thanks for the comment. We have rewritten the introduction section based on your suggestions.

2.3) In the substrate scope of this work, except for a few examples with *er* >95:5, most products were formed in 60-80% *ee* only. Also, the ansa chain length is limited to 13-15 for a reasonable enantioselectivity, which is more limited than the previous CPA-DKR work (>90% *ee* for ansa chain length 12-19). I think further improvement on the enantioselectivity and substrate generality should be achieved for this work to be published in *Nat. Commun.*

Our response:

Thanks for the comments. We have furtherly optimized the reaction conditions based on your suggestions. The use of 2,4,6-Cy₃C₆H₂ substituted indanol-derived NHC catalyst and increase of reaction temperature led to a higher enantioselectivity, which might arise from a faster racemization rate of the substrate. Under the new reaction conditions, most of the products were obtained with er \geq 95:5.

Reviewers' Comments:

Reviewer #1:

Remarks to the Author:

The new version of the manuscript addresses the previous deficiencies and criticisms. It is necessary to improve the writing in some of the new paragraphs.

Reviewer #2:

Remarks to the Author:

In the revised manuscript, the authors have addressed my concerns nicely, especially that they were able to boost the enantioselectivity for many products to about 90% ee range. I am happy to support acceptance of this work to publication in Nat. Commun.

1. Responses to reviewer 1:

1.1 The new version of the manuscript addresses the previous deficiencies and criticisms. It is necessary to improve the writing in some of the new paragraphs.

Our response:

We are pleased to thank this reviewer to give this positive response, we have proof read the manuscript and made necessary corrections of the new paragraphs.

2. Responses to reviewer 2:

2.1 In the revised manuscript, the authors have addressed my concerns nicely, especially that they were able to boost the enantioselectivity for many products to about 90% ee range. I am happy to support acceptance of this work to publication in Nat. Commun.

Our response: We are pleased to thank this reviewer to give this positive response.

Should you have any further questions, please feel free to contact me. We are looking forward to hearing from you.